# Synergistic Effects of Anionic/Cationic Dendrimers and Levofloxacin on Antibacterial Activities

**DOI:** 10.3390/molecules24162894

**Published:** 2019-08-09

**Authors:** Natalia Wrońska, Jean Pierre Majoral, Dietmar Appelhans, Maria Bryszewska, Katarzyna Lisowska

**Affiliations:** 1Department of Industrial Microbiology and Biotechnology, Faculty of Biology and Environmental Protection, University of Lodz, 12/16 Banacha Street, 90-236 Lodz, Poland; 2CNRS, LCC (Laboratoire de Chimie de Coordination), 205 route de Narbonne, BP 44099, F-31077 Toulouse CEDEX 4, France; 3Leibniz Institute of Polymer Research Dresden, Hohe Straße 6, D-01069 Dresden, Germany; 4Department of General Biophysics, Faculty of Biology and Environmental Protection, University of Lodz, 141/143 Pomorska Street, 90-236 Lodz, Poland

**Keywords:** dendrimers, antibacterial activity, levofloxacin, synergistic effects

## Abstract

Despite the numerous studies on dendrimers for biomedical applications, the antibacterial activity of anionic phosphorus dendrimers has not been explored. In our research, we evaluated the antibacterial activity of modified polycationic and polyanionic dendrimers in combination with levofloxacin (LVFX) against Gram-negative (*Escherichia coli* ATCC 25922, *Proteus hauseri* ATCC 15442) and Gram-positive (*Staphylococcus aureus* ATCC 6538) bacteria. In the case of Gram-negative bacteria, we concluded that a combination of dendrimers and antibiotic gave satisfactory results due to a synergistic effect. The use of fluoroquinolone antibiotics, such as LVFX, not only caused resistance in disease-causing microorganisms but also increased environmental pollution. Therefore, reduction of drug dosage is of general interest.

## 1. Introduction

Antimicrobial resistance is an increasing global problem. Antimicrobial drugs, antibiotics, often cause bacterial resistance. This is responsible for many therapy failures. As a consequence, it is mandatory to develop new strategies leading to original antimicrobial compounds. Thus, the involvement of dendrimers can pave the way for new therapeutic approaches. Dendrimers are soft hyperbranched nanoparticles whose size, topology, and flexibility can be rigorously controlled during their synthesis. They are generally prepared via the reiteration of a sequence of reactions starting from a core molecule to introduce repeating branching units. Finally, this allows for the grafting of a variety of functional groups on their outer shell which can interact with other (macro) molecules [1,2,3,4]. In fact, dendrimers can act as active species per se or as cargo for drugs. The unique structure of dendrimers allows for their application as ideal carriers of imaging or transfection agents, using various ways of administration [5,6,7,8]. Numerous studies have also presented dendrimers as complexing agents for antimicrobial drugs [9,10,11,12]. Dendrimers, as drug carriers, smoothly improve the solubility and stabilization of the drug in blood [13,14,15,16]. Additionally, cationic dendrimers are promising candidates for the development of antibacterial drugs. Many studies describe the antimicrobial activity of poly(amidoamine) (PAMAM) and poly(propyleneimine) (PPI) dendrimers [17,18,19,20]. Ortega et al. [21] presented the antibacterial potential of new ammonium and amine-terminated carbosilane dendrimers against Gram-negative and Gram-positive bacteria. Moreover, Cheng et al. [22] observed an improvement in the solubility and activity of fluoroquinolone (FQ) antibiotics after adding the PAMAM dendrimers to the solution of FQ.

However, the main limitation of the use of polycationic dendrimers in medicine is their cytotoxicity. Cationic dendrimers can be toxic to eukaryotic cells [4,20,23,24]. Gholami et al. [20] demonstrated that the cytotoxicity effect of the PAMAM-G7 dendrimer on the HCT 116 and NIH 3 T3 cells depends on the concentration of the compounds and the exposure time. Cytotoxicity mechanisms include disruption of the cell membrane’s integrity and rise in the intracellular level of reactive oxygen species (ROS), which could induce apoptosis [25]. However, a modification of the dendrimer surface may reduce their cytotoxicity. In a previous study, we examined the influence of maltose modifications on the properties of PPI dendrimers [19]. In this case, open-shell (PPI dendrimers possessing 25% modified amino groups due to the attachment of maltose groups) and dense-shell (PPI dendrimers possessing mainly 100% modified amino groups due to the attachment of two maltose groups) PPI glycodendrimers were almost non-toxic toward four different cell lines (B14, HepG2, N2a, BRL-3A) at a concentration of 1 µM (cell viability was higher than 90%). On the other hand, 3 µM concentration of their counterpart, unmodified PPI dendrimer, led to a rapid decrease in cell viability (50%) in B14 and N2a cells. Similarly, open-shell PPI glycodendrimers were more toxic than those with dense-shell glycoarchitecture [19]. In marked contrast, anionic dendrimers showed no or very low toxicity [26].

Because of many interesting properties, phosphorus dendrimers are of great interest for drug delivery applications, anti-prion, imaging [27], anti-cancer [28,29,30], and anti-inflammatory agents [31,32,33]. Moreover, polyanionic and polycationic phosphorus dendrimers have been proposed as carriers for photosensitizers in photodynamic therapy [34]. Thus far, there is little information about the biological properties of anionic dendrimers. Furthermore, literature data have not focused on the antibacterial activity of anionic phosphorus dendrimers.

The main purpose of our study was to examine whether the co-administration of modified polycationic or polyanionic dendrimers and LVFX (an antibiotic of the FQ family) would modify the antimicrobial activity of LVFX and allow for reductions in the doses of the drug. We chose LVFX because most Gram-negative bacteria have already developed resistance to FQ [35,36]. Indeed, FQ, a popular group of antibiotics, are widely used in the treatment of humans and animals [37,38,39]. Piperazine substitution at the C-7 position has resulted in broad coverage of clinically useful FQ antibacterial agents like LVFX, ciprofloxacin, and norfloxacin [40,41]. LVFX, a third-generation FQ, has been reported to be active against Gram-negative and Gram-positive bacteria [42]. The mechanism of action of these antibiotics involves the inhibition of bacterial DNA replication, transcription, reparation (repaired), and recombination by interaction with DNA-topoisomerase IV and DNA gyrase [43,44]. In Europe and the USA, LVFX is used for the treatment of respiratory tract infections, genitourinary infections, acute pyelonephritis, chronic prostatitis, post-inhalational anthrax, and different skin infections [45,46]. Excessive use of this antibiotic not only leads to increased bacterial resistance but also contaminates the environment. There are many reports on LVFX contamination of aquatic environments [47,48,49]. This contamination mainly comes from hospitals and pharmaceutical industries. Therefore, striving to reduce the doses of LVFX in therapy is very important.

In this study, we report the antimicrobial activity of LVFX (Figure 1) in combination with PPI glycodendrimers with dense maltose shell (PPI-G3-DS-Mal) (Figure 2) or an anionic phosphorus dendrimer (AN G4) (Figure 3) on *E. coli* ATCC 25922, *P. hauseri* ATCC 15442, and *S. aureus* ATCC 6538. *Escherichia coli* is the most commonly isolated uropathogen. Also, Proteus causes complicated urinary infections accompanied by formation of urinary stones [50]. Gram-positive strain *S. aureus* often causes skin diseases, heart valves, and bones infections [51].

## 2. Results and Discussion

For the first time, we estimated the combined action of PPI glycodendrimer PPI-G3-DS-Mal and LVFX against the common human pathogens, *E. coli*, *P. hauseri*, and *S. aureus*. The knowledge of the concentration-dependent non-toxicity of the third-generation glycodendrimer PPI-G3-DS-Mal (Figure 2) toward different cell lines [19,52,53] enabled us to explore their biological action against bacteria cells. PPI-G3-DS-Mal dendrimer is non-toxic to bacterial cells also, and when applied alone, does not exhibit bacteriostatic activity to *E. coli*, *P. hauseri*, and *S. aureus* (Figure 4, Figure 5 and Figure 6). However, the co-administration of PPI-G3-DS-Mal dendrimer and LVFX considerably reduces the growth of *E. coli*, while it did not enhance the antimicrobial activity in the case of *S. aureus,* and it had slight effect on *P. hauseri*. The addition of PPI-G3-DS-Mal dendrimer (10 μM) and LVFX (0.01 μg/mL) limits *E. coli* growth by 80%, whereas, when the dendrimer or antibiotic alone is applied, the growth is inhibited by only 6% or 4%, respectively (Figure 4). The simultaneous addition of LVFX and PPI-G3-DS-Mal dendrimer to a culture of *P. hauseri* limits the growth of bacteria by only 10%–15% (of LVFX as reference) at all tested concentrations (Figure 5). We can conclude that the combination of this dendrimer and antibiotic gives satisfactory results due to a synergistic effect in the case of *E. coli*. Our previous results showed that the PPI-G3-DS-Mal dendrimer enhanced the antibacterial activity of amoxicillin against Gram-negative microorganisms [54]. However, it should be remembered that those antibiotics have distinct mechanisms of action.

We also investigated the antimicrobial effect of the tested compounds on the Gram-positive strain, *S. aureus*. We observed that the simultaneous addition of LVFX and PPI-G3-DS-Mal does not strengthen the antimicrobial effect when compared to the effect on the Gram-negative bacteria, *E. coli*, on which it shows five times greater inhibition of bacterial growth (compared to the antibiotic alone). Moreover, in the case of *S. aureus* culture (LVFX+ PPI-G3-DS-Mal), we notice the abolition of the antibacterial effect of LVFX (Figure 6). The sensitivity of Gram-negative and Gram-positive bacteria to the tested dendrimer may be dependent of their cell wall construction or mechanism of interactions between the membrane and the dendrimer.

The mechanism of antibacterial action of dendrimers is mainly related to their effect on the permeability of the cellular surface of microorganisms. Based on our previous research, we know that the parental third-generation PPI dendrimer itself and the open-shell third generation PPI glycodendrimer, with 25% maltose modification of the terminal amino groups, exert detrimental effects on bacteria cell membrane [19]. We also observed an increase in the permeability of the *P. aeruginosa* cell membrane after incubation with different PPI dendrimers [54]. In this study, for the first time, we report that the PPI glycodendrimer PPI-G3-DS-Mal (Figure 2) has the potential to enhance antibacterial activity against Gram-negative bacteria when combined with FQ antimicrobials. This is very important because infections caused by the Gram-negative strain cause serious problems [55]. The resistance of Gram-negative bacteria is associated with the permeability of their outer membrane. This limits the access of drugs to their intracellular targets. As a consequence, infections caused by these bacteria are complicated to treat.

In contrast, very little is known about the antibacterial properties of anionic dendrimers. It is assumed that anionic dendrimers may exhibit antimicrobial activity with minimal eukaryotic cell cytotoxicity due to their general non-cytotoxicity, and low toxicity has been observed in whole zebrafish animal development studies [56]. Moreover, amphiphilic compounds act through perturbation and disruption of the prokaryotic membrane. Meyers et al. observed the antibacterial activity of anionic amphiphilic dendrimers against the Gram-positive strain, *Bacillus subtilis* AG174, and noted minimal eukaryotic cell toxicity [57].

We found that the addition of anionic phosphorus dendrimers enhances the antibacterial activity of LVFX against *E. coli*, particularly at higher concentrations of the antibiotic (Figure 7). The co-administration of anionic phosphorus dendrimers at 10 μM concentration with 0.05 μg/mL of LVFX reduces bacterial growth by 70%, whereas for the antibiotic applied alone, the inhibition is around 15%. The anionic phosphorus dendrimers (10 and 50 μM) did not inhibit the growth of *E. coli*. In the case of *P. hauseri*, the results obtained are not as satisfactory (Figure 8) as for *E. coli* (Figure 7). In this strain, co-administration of the anionic phosphorus dendrimer AN G4 (10, 50 µM) with higher doses of LVFX (0.5 µg/mL) increases the antibacterial effect by 10–15%. The antibacterial activity of the anionic phosphorus dendrimer and/or LVFX was also investigated for the Gram-positive strain. However, no significant effect of these compounds on the inhibition of *S. aureus* growth was noted (Figure 9). Interestingly, in the *S. aureus* culture containing LVFX and dendrimers (PPI-G3-DS-Mal or AN G4), the abolition of the effect of action is observed. This biological action for anionic phosphorus dendrimer as a potential antimicrobial drug has not been described yet.

## 3. Materials and Methods

### 3.1. Reagents

Unmodified PPI (fourth-generation with 32 amino groups, MM 3514 g/mol) dendrimer was purchased from SyMO-Chem (Eindhoven, The Netherlands). To compare it with other polyamino dendrimers, we followed the nomenclature of PPI dendrimers by Tomalia et al. [58]. The referenced generation of PPI dendrimers from literature [19,52] is also considered by the nomenclature suggestion of Tomalia et al. [58]. Thus, the PPI dendrimer is of the third generation. The synthesis of a maltose-modified third-generation PPI dendrimer (PPI-G3-DS-Mal, molecular weight 24,397 g/mol) was performed at Leibniz Institute of Polymer Research Dresden, Germany, following the process of synthesis and characterization described by Klajnert et al. [52]. The degree of maltose substitution (95%–100%) in PPI-G3-DS-Mal (Figure 2) was determined using 1H NMR [52]. LVFX was purchased from Sigma-Aldrich (Hamburg, Germany). The anionic phosphorus dendrimer AN-G4 was synthesized in Laboratoire de Chimie de Coordination, CNRS, Toulouse, France. The anionic phosphorus dendrimer AN-G4 was prepared according to the method previously reported [59].

### 3.2. Determination of Antimicrobial Activity

The antimicrobial activity of the dendrimers and the antibiotic was examined using a modified broth microdilution method as per standards defined by the National Committee for Clinical Laboratory Standards (NCCLS M07-A8). The analysis involved incubating *P. hauseri* (ATCC 15442), *E. coli* (ATCC-25922), and *S. aureus* (ATCC 6538) with serial dilutions of the compounds on microtiter plates and measuring cell density (OD) spectrometrically at 620 nm. The antimicrobial activity of the dendrimers was determined at 10 and 50 μM for the maltose-modified and anionic phosphorus dendrimers. The germicidal activity of LVFX was measured at different concentrations against the two bacterial strains at 0.7–5.0 µg/mL (see Results). The data were compared to three independent controls: (i) bacteria incubated in the medium, (ii) bacteria incubated with dendrimers, and (iii) bacteria incubated with antibiotics.

## 4. Conclusions

Our study showed that the third-generation PPI glycodendrimer with PPI-G3-DS-Mal and the anionic phosphorus dendrimer of the fourth generation, bearing 96 terminal carboxylate groups, strengthen the antibacterial activity of LVFX, allowing for a reduced antibiotic dose for bacterial treatment. This is important because of the widespread use of antibiotics, which leads to drug resistance among microorganisms, contributing to increasing environmental pollution. Moreover, this is the first attempt to use anionic phosphorus dendrimers alone or with LVFX as antimicrobial agents. The mechanism of action of the synergistic effect of the studied compounds on microbial cells will be a subject of study in the future.

## Figures and Tables

**Figure 1 molecules-24-02894-f001:**
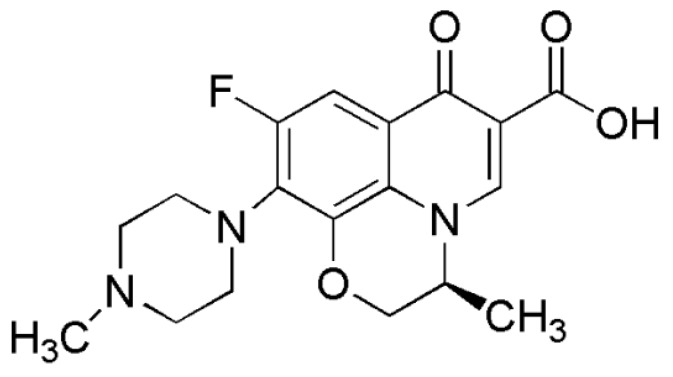
Chemical structure of levofloxacin (LVFX).

**Figure 2 molecules-24-02894-f002:**
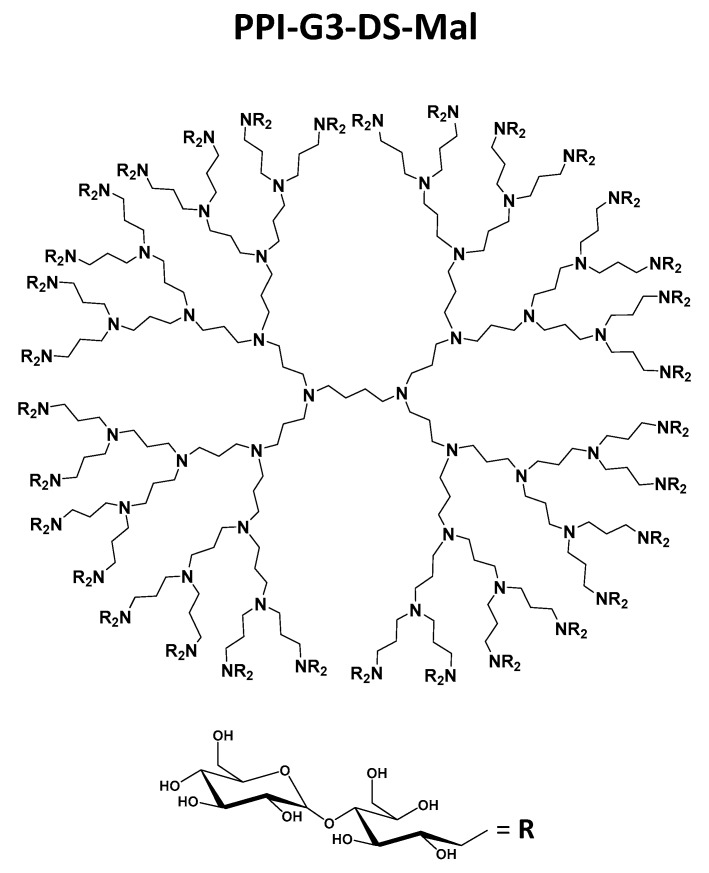
Simplified structure of third generation poly(propyleneimine) (PPI) dendrimer with dense maltose shell (PPI-G3-DS-Mal) at which each terminal amino group has preferably attached two maltose units.

**Figure 3 molecules-24-02894-f003:**
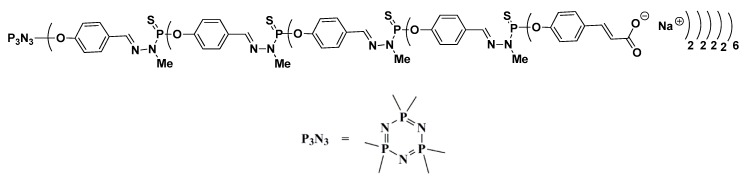
Structure of polyanionic phosphorus dendrimers (AN G4).

**Figure 4 molecules-24-02894-f004:**
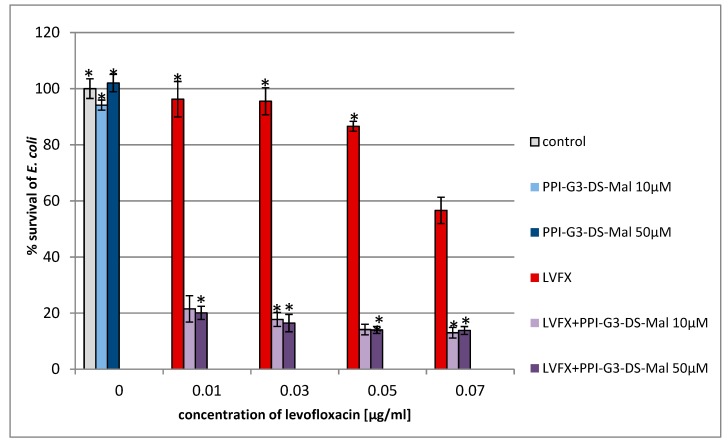
Growth of *E. coli* after 24 h of incubation with PPI dendrimer, PPI-G3-DS-Mal, and/or LVFX. Each bar represents an average and SD taken from n ≥ 3 wells from three independent experiments. The comparison was made using one-way analysis of Student’s t-test. * *p* ˂ 0.05 vs. control group.

**Figure 5 molecules-24-02894-f005:**
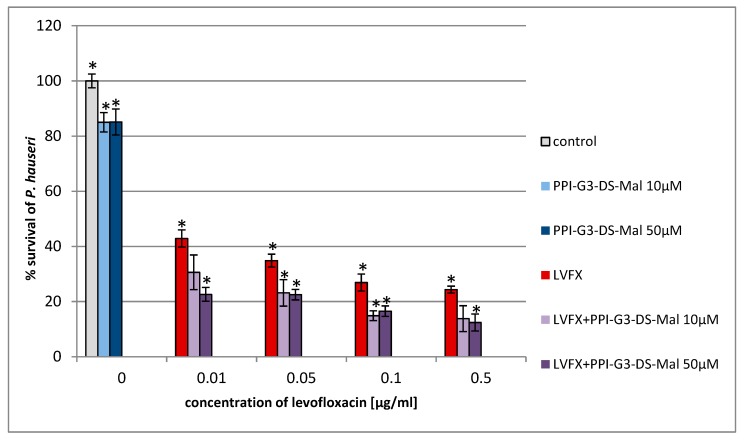
Growth of *P. hauseri* after 24 h of incubation with poly(propylene imine) dendrimer PPI-G3-DS-Mal and/or LVFX. Each bar represents an average and SD taken from n ≥ 3 wells from three independent experiments. The comparison was made using one-way analysis of Student’s t-test. * *p* ˂ 0.05 vs. control group.

**Figure 6 molecules-24-02894-f006:**
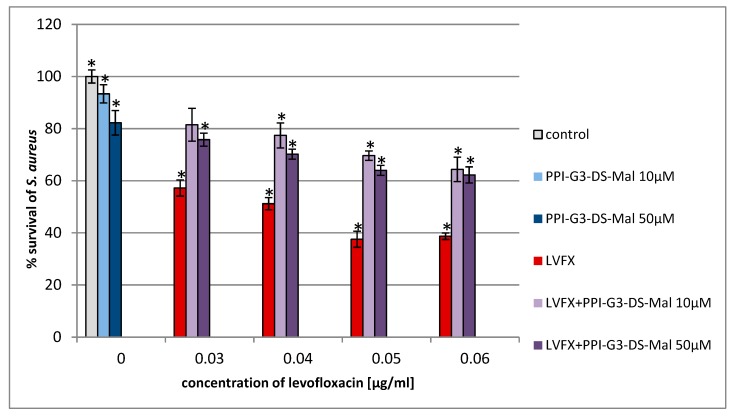
Growth of *S. aureus* after 24 h of incubation with poly(propylene imine) dendrimer PPI-G3-DS-Mal and/or LVFX. Each bar represents an average and SD taken from n ≥ 3 wells from three independent experiments. The comparison was made using one-way analysis of Student’s t-test. * *p* ˂ 0.05 vs. control group.

**Figure 7 molecules-24-02894-f007:**
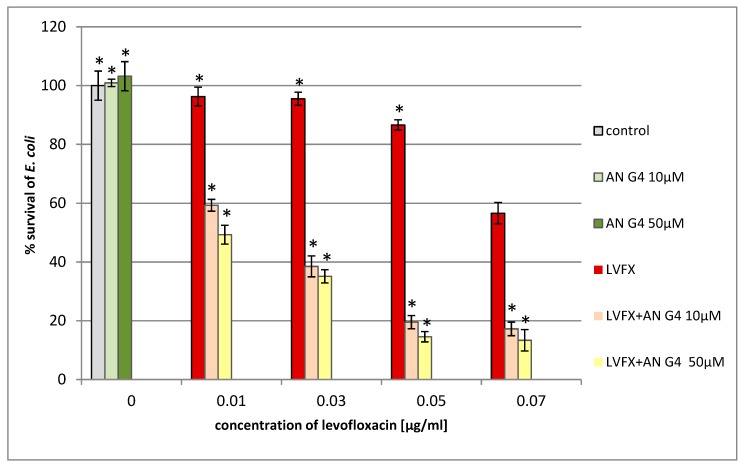
Growth of *E. coli* after 24 h of incubation with anionic phosphorus dendrimer AN G4 and/or LVFX. Each bar represents an average and SD taken from n ≥ 3 wells from three independent experiments. The comparison was made using one-way analysis of Student’s t-test. * *p* ˂ 0.05 vs. control group.

**Figure 8 molecules-24-02894-f008:**
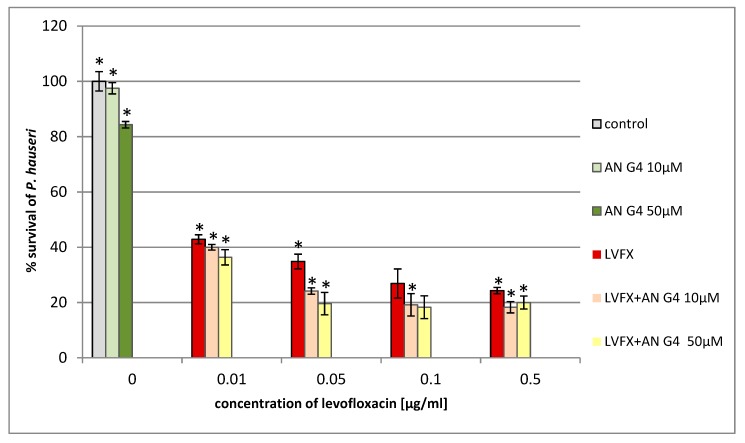
Growth of *P. hauseri* after 24 h of incubation with anionic phosphorus dendrimer AN G4 and/or LVFX. Each bar represents an average and SD taken from n ≥ 3 wells from three independent experiments. The comparison was made using one-way analysis of Student’s t-test. * *p* ˂ 0.05 vs. control group.

**Figure 9 molecules-24-02894-f009:**
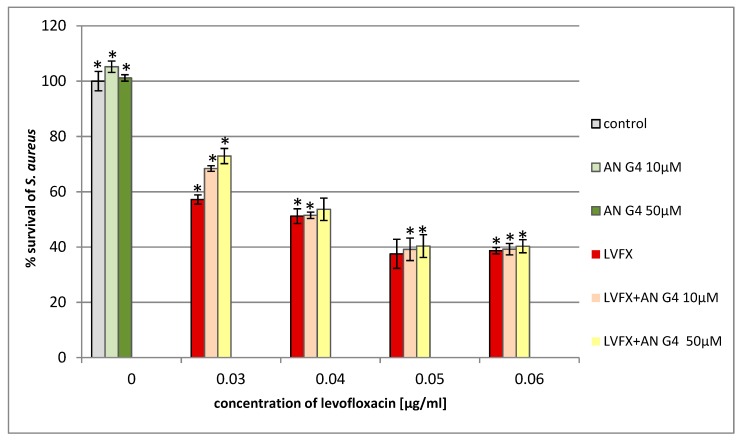
Growth of *S. aureus* after 24 h of incubation with anionic phosphorus dendrimer AN G4 and/or LVFX. Each bar represents an average and SD taken from n ≥ 3 wells from three independent experiments. The comparison was made using one-way analysis of Student’s t-test. * *p* ˂ 0.05 vs. control group.

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
