# Peer review of "Synergistic Effects of Anionic/Cationic Dendrimers and Levofloxacin on Antibacterial Activities"

_molecules, 2019, doi:10.3390/molecules24162894_

Round 1

Reviewer 1 Report

This manuscript by Lisowska studies the synergistic effects of cationic/anionic dendrimers and levofloxacin on antibacterial activities and the results show that the combinations of levofloxacin and PPI glycodendrimer or phosphorus dendrimers reduce the growth of E. coli and P. hauseri but show opposite effect on S. aureus. The work appears to have not been reported before. Overall the manuscript was well organized, and the full story was nicely delivered. The following items should be addressed prior to publication:

1.     Consider another title: Synergistic Effects of Anionic/Cationic Dendrimers and Levofloxacin on Antibacterial Activities.

2.     Reduce the size of structure of LVFX in Figure 1. It is unnecessarily too big.

3.     The first paragraph at the 2. Results and Discussion should be deleted. The reason why LVFX was chosen for the study was mentioned at the Introduction, and thus not needed to be duplicated at the Results.

4.     In Figures 4-9, use different colors (e.g. red, blue, purple… whatever colors authors like) to represent the control and various combinations of dendrimer and LVFX rather than different gray levels, which is less readable.   

5.     The statement at line 165-166 at the page 6/12 was incorrect. Data at Figure 7 showed that dendrimer applied alone doesn't inhibit the E. coli.

6.     At end of line 171, did the author mean S. aureus rather than P. hauseri?!!

Reviewer 2 Report

This manuscript describes antimicrobial activity of levofloxacin (LVFX) in the presence of PPI dendrimer with dense maltose shell (PPI-G3-DS-Mal) or an anionic phosphorus dendrimer (AN-G4). The authors found the combination with LVFX and PPI-G3-DS-Mal inhibited significantly the growth of E. coli. and Gram-positive bacteria, as compared with the case using LVFX alone. The authors also found that the combination with LVFX and AN-G4 reduced growth of E. coli. and Gram-negative bacteria. The strong synergistic effects were found in some combination. The finding may allow us to reduce dose of LVFX in the treatment bacterial infections, which will contribute to prevent development of the drug resistance bacteria. The work have been done carefully and the manuscript is written in a clear and concise manner. Therefore, I would recommend the manuscript be published in Molecules. I would recommend that in Figure 3, the structure of the central core for the dendrimer should be shown more clearly instead of P3N3.

Author Response

Response to the Reviewer:

Thank you for a nice review. The Figure 3 has been corrected according to the Reviewer’s suggestion. Now, the structure of the dendrimer is shown more clearly.

I hope that we have satisfactorily responded to the concerns of the Reviewers.

Reviewer 3 Report

The manuscript describes the effect of a combination of dendrimers with an antibiotic (LVFX) on antibacterial activity. The synthesis and the antibacterial activity of the reported glyco-dendrimer is known [19, 52, 54]. This study reports the effect of a combination of the dendrimer with LVFX.

The authors should include details on synthesis and analytical data for the anionic phosphorous dendrimer.

Moreover some sentences should be rephrased:

Lines 111-113 “co-administration of PPI-G3-DS-Mal dendrimer and LVFX considerably reduces the growth of tested bacteria.”  Co- administration considerably reduces only the growth of E. coli, while it did not enhance the antimicrobial activity in the case of S. aureus and it had slight higher effect on P. hauseri.

Conclusion: The effect of a combination of glycol-dendrimer and antibiotic does not seem to be synergistic in all cases. Synergistic effect is pronounced in the case of E. coli.

Lines 103-104: This was already mentioned in lines 82-83
